# Analysing Pine Disease Spread Using Random Point Process by Remote Sensing of a Forest Stand

Rostyslav Kosarevych [1,2,*], Izabela Jonek-Kowalska [3], Bohdan Rusyn [1,4], Anatoliy Sachenko [4,5] and Oleksiy Lutsyk [1]

1 Karpenko Physics and Mechanics Institute, National Academy of Sciences of Ukraine, 5 Naukova Str., 79060 Lviv, Ukraine; b.rusyn.prof@gmail.com (B.R.); olutsyk@yahoo.com (O.L.)
2 Department of Artificial Intelligence Systems, Lviv Polytechnic National University, 12 Stepan Bandera Str., 79000 Lviv, Ukraine
3 Department of Economics and Informatics, Silesian University of Technology, 2A, Akademicka Str., 44-100 Gliwice, Poland; izabela.jonek-kowalska@polsl.pl
4 Department of Informatics and Teleinformatics, Kazimierz Pulaski University of Technology and Humanities in Radom, 29, Malczewskiego Str., 26-600 Radom, Poland; sachenkoa@yahoo.com
5 Research Institute for Intelligent Computer Systems, West Ukrainian National University, 11 Lvivska Str., 46009 Ternopil, Ukraine
* Correspondence: r.y.kosarevych@gmail.com

**Abstract:** The application of a process model to investigate pine tree infestation caused by bark beetles is discussed. The analysis of this disease was carried out using spatial and spatio−temporal models of random point patterns. Spatial point patterns were constructed for remote sensing images of pine trees damaged by the apical bark beetle. The method of random point processes was used for their analysis. A number of known models of point pattern processes with pairwise interaction were fitted to actual data. The best model to describe the real data was chosen using the Akaike information index. The residual K−function was used to check the fit of the model to the real data. According to values of the Akaike information criterion and the residual K−function, two models were found to correspond best to the investigated data. These are the generalized Geyer model of the point process of saturation and the pair interaction process with the piecewise constant potential of a pair of points. For the first time, a spatio−temporal model of the contagious process was used for analysis of tree damage.

**Keywords:** pine disease; infestation; spatial modelling; random point process

## 1. Introduction

Studies of pine diseases causing desiccation, in particular caused by bark beetles, have been conducted for a considerable time and are based on various approaches [1–8]. Biologists mainly use field research to analyse damaged trees, pests and pathogen impacts in certain regions over several years [9,10]. Such research involves choosing an experimental forest area without infected trees and then observing the expansion of the disease over time [9–11]. These studies are valuable for their thoroughness, as they allow the collection of diverse data. The research duration is constrained by the development cycles of the general environment and the specifics of the investigated area and pests. However, we should also consider that the mean values of other parameters, for example, of the external environment, such as temperature, amount of precipitation, etc., can change significantly, which is an important factor affecting the formation of dependencies between diseases and the environment [12].

Due to the seriousness of the problem, alternative approaches are used. In particular, remote sensing of forest areas can address the problem on a larger scale but is limited in its accuracy in determining the cause of the drying of a specific type of stand [13–16]. Surveys

based on remote sensing, while requiring less effort, can cover considerably larger areas, hence producing data that are more comprehensive. Moreover, monitoring the disease over some time allows for describing the nature and pace of its spread over a large area [17–19]. Combining field and remote studies seems ideal, as it will allow us to aggregate the most of relevant information necessary for understanding the nature of the course of the disease. Both approaches involve disease analysis and aim to gain a better understanding of how the disease initiates and spreads as well as identifying factors which influence its progression. Such information is essential in developing solutions for disease control and elimination. Unfortunately, such combined studies are rarely found in the literature.

The analysis of the disease phenomenon as well as its onset and spread intends to identify core processes that take place and to formally describe them. As a rule, such a description is given by a statistical model of the process reflecting the development of the disease in the case of a single tree [20] or its local distribution [21].

Modelling disease expansion constitutes a significant part of the research into the phenomenon [2,6,21–25]. The majority of studies have looked into predicting the quantitative assessment of disease distribution [2,21,24–26]. Much less attention has been paid to the spatial distribution of the contamination among trees in finite regions [7,11,17,18,23]. While the former approach has more practical application by allowing one to estimate the amount of wood loss, the latter allows also an assessment of disease connection to other natural phenomena, for example, storms and fires [11].

A cluster model of tree death was studied in [24]. After testing the hypothesis about the complete spatial randomness of the location of dead trees and rejecting it based on known criteria [23], the authors, using the spatial statistics presented in [27], determined and investigated the statistical characteristics of cluster accumulation of dead trees.

Paper [5] analyses spatio−temporal patterns of the death of trees in the China Camp State Park, CA, USA, obtained over a period of four years via remote sensing images. The simulations using random point processes made it possible to establish dependencies between dead trees of the same species and the relationship between dead and infected trees.

In paper [11], the spatial superposition of infection and falling of trees due to storms is analysed based on point patterns of the region in central Switzerland. The characteristics of the neighbourhood of trees damaged by bark beetles and storms were calculated. Infestations by bark beetles were found to form clusters over short distances (<500 m). In contrast, the spatial distribution of dead trees from storms followed a non−uniform Poisson distribution as a consequence of ecological covariates. It is shown that the factors that positively affect the probability of tree destruction due to a storm and infection with bark beetles are the height of the topographical relief, the presence of slopes and significant stands of Norway spruce.

It is convenient to use models based on spatial random point processes for describing and analysing the properties of many physical processes, in particular plant ecology [28]. They operate with objects that can be described as random configurations of points on a plane or space. Each issue of such a configuration can also reflect a particular event. Based on the analysis of their mutual location, namely the distances between the nearest neighbours, or the number of objects in a specific limited area, it is possible to establish the underlying processes that lead to the formation of such configurations. We believe that the process of disease spread in pine plantations can be described by spatial models of infectious diseases. These models can be represented graphically by random point processes with interactions.

As mentioned earlier, a large number of models have been used to describe the course of plant diseases. However, random point processes are not widely used in these models. For researchers studying models of such processes, it is largely due to the unsatisfied condition of the randomness of the location of the event—the object of research. Strictly speaking, since the establishment of plantations is fixed, it is not entirely correct to treat such areas as a random configuration of points marking damaged trees. As shown in [29], tree locations are fixed covariance values.

However, there are known studies dedicated to the analysis of another type of tree damage, namely forest fires [30]. In [31,32], techniques based on spatio−temporal processes [33] were developed, describing patterns of forest fires. To overcome the dependence of fires on the fixed location of plantations, point configuration was constructed based on the location of the centres of the areas of recorded forest fires. By choosing the centre of the fire zone as an element of the point image, we remove the fixed position of the event, since now one event is considered a set of trees that were destroyed by fire at the same time and their number is random.

We aimed to apply a similar approach to the formation of random point configurations to describe the occurrence of plant diseases. It is fairly logical to use the method of spatial or spatio−temporal point processes for their analysis. One of the ways to generate the random point configurations of damaged trees is by analysis of remote sensing images [17–19]. Suppose we divide an image into patches and mark those where a particular threshold value of spectral characteristics has been reached in either the fragment elements or their generalizations. This way a random set of fragment centres and a random point image will be formed.

## 2. Materials and Methods

A spatial point pattern is a set of locations, unevenly distributed in the study region U, where events were recorded, such as the location of trees in the natural forest. The spatial point pattern is usually modelled as the realization of a spatial stochastic process described by a set of random variables: $N(U_g)$, $U_g \subset U$, where $N(U_g)$ is the count of events appearing in the $U_g$ subregion [34].

The inhomogeneous Poisson random point process is a widely accepted model for describing the processes accompanying the spread of infectious and contagious diseases [11]. This Poisson process is a form of the uniform Poisson process, with the intensity $\lambda$ to vary by a non−homogeneous intensity function $\lambda(u)$, which is estimated using the probability density function [30]. Despite its wide application, the inhomogeneous Poisson process has limited capabilities in describing random events because it is derived via "thinning" of the homogeneous Poisson process. First, a uniform Poisson process of particular intensity $\lambda$ is generated. Next, randomly and independently, every individual point is either eliminated or preserved with probability proportional to the intensity of the process at this point. This approach has limited possibilities for implementing different types of point configurations.

### 2.1. Spatial Modelling Using Interaction Processes

The above modelling based on a wide range of random Gibbs processes was studied to improve accuracy in simulating the spread of the disease in pine plantings. It comprises a large family of models describing interactions of various types between events and includes modelling of regular point patterns, point patterns with aggregation of events as well as their combinations. Gibbs processes were introduced in statistical physics to describe systems of particles interacting and are defined using the density function relative to the Poisson process [29].

The probability of an arbitrary finite configuration of points is determined by the local interaction and depends on the interaction with neighbours, which can be defined in various ways. If only one neighbour is considered, then the Gibbs process is a model of pair interactions, with the density function given by the expression [35]:

$$f(u_1, \dots u_n) = \alpha \prod_i \beta(u_i) \prod_{i<j} \gamma(u_i, u_j), \tag{1}$$

where $u_1, \dots, u_n$ describes the pattern points. In this expression, the first product passes through all pattern points, and the second product passes through all pairs of pattern points. Each point $u_i$ of the image adds a coefficient $\beta(u_i)$ to the probability density function, and each pair of points $u_i$, $u_j$ supplies a coefficient $\gamma(u_i, u_j)$, respectively. The intensity of the process is given by $\beta(u)$ while the function $\gamma(u,v)$ reflects the properties of the second order

and is called the pairwise interaction function. For models of pair interaction, this function is assumed to be symmetric and isotropic.

There is a wide range of known models of point processes with pair interactions that allow building descriptions of natural processes. Some random spatial point−process models are realized by the "spatstat" package of the R software environment (version 4.3.1) [30]. By selecting relevant parameters, the model was adjusted to accurately reflect the actual process describing the studied point pattern based on factual point images of tree diseases. Available models allow the use of spatial covariates that can represent information about the landscape of the area, such as the level of groundwater, the location of bird or insect nests, etc. The use of such additional information certainly increases the accuracy of the model; however, it is not always available when using remote sensing data. It is either missing in the studied area or obtained as a result of the assessment. An analysis of the literature [9–11] shows the predominant influence of soil moisture on the course of tree infection. Hence, as spatial covariates, we used spatially correlated random fields of soil moisture. These fields were generated from the intensity values of healthy trees in the study area.

Even if we obtained a point process model that fits well to a set of spatial point patterns, its correspondence to the data still has to be verified since it does not necessarily have to be the same as the actual data pattern. One of the ways to evaluate the conformity of the model to real data is via the well−recognized Akaike information criterion (AIC), whereby the smaller the AIC value, the better model corresponds to the entire process. The Akaike Information Criterion is calculated as:

$$\text{AIC} = -2\log L + k \times p, \tag{2}$$

where L is the maximum likelihood of the fitted model and p is the model complexity penalty, which is mostly equal to the numbers of degrees of freedom in the model [35].

### 2.2. Infection and Contagion as Spatio-Temporal Processes

For specific spatial processes, the temporal component may also be considered and taken into account when modelling the phenomenon of interest (e.g., disease distribution or air pollution risk assessment). In such cases, spatio−temporal point processes should be considered instead of purely spatial ones when choosing a suitable model. Data analysis of point processes in time is broadly covered in the literature [33,35,36], but less so when it comes to general analysis of spatio−temporal processes [34,37,38].

The spatio−temporal process events compose a set of points P = {($u_i$,$t_i$): i = 1, 2, . . .}, where $u_i$ denotes the place and $t_i$ the time of the $i^{th}$ event. Such data are available for analysis as points ($u_i$,$t_i$): i = 1,. . .; n, which form a partial realization of the process limited by the bounded space–time area of observation U × T, where U is a spatial region, and T is time interval.

As in the case of spatial processes, the main characteristics of spatio−temporal processes are the intensities of the first and second orders: λ(u,t) and λ(($u_i$,$t_i$), ($u_j$,$t_j$)) accordingly. The first is defined as the limit of the ratio of the expected number of events that occurred in a particular region during a specific time interval to the size of the area and the duration of the time interval, provided that the area of the region and the duration of the interval approach zero. The relationships between the pair of events in U×T are determined by the second−order intensity as the marginal ratio of the expected number of events in a pair of regions to the areas of these regions under the condition that their areas approach zero [35]. Taking into account the temporal position provides a natural ordering of the points of a random process, which would be impossible for a purely spatial process. It also allows one to visualize the development of the process over time.

In the case of a wide spread of harmful insects and diseases, tree death spreads intensively; thus, it is reasonable to assume that the processes of disease are either infectious or contagious. Infectious diseases, as a rule, have an incubation period during which outward signs of the disease are invisible. Although we are convinced that the drying of

pine trees is due to infection of the trees by bark beetles, the results are noticeable when the trees are contagious. In addition, a tree can successfully fight infection [39]. Thus, we are interested in modelling the spreading of the contamination of trees, not their infection.

Tools required for analysis of such processes are available in the "stpp" package of the R software environment that allows constructing, modelling and analysing spatio−temporal point images [40]. We believe that the spreading process of pine drying is a process of infection. Algorithmically, a simple disease process is given in [40] as a sequence of steps by which a set of events is formed, each of which is uniformly distributed in a region that is the intersection of the entire study area and a circle of a certain radius centred at the location of the previous event. The time of occurrence of each event is also a random variable, uniformly distributed over a time interval of a certain duration from the moment of occurrence of the previous event. The functions that figure out the infection process are determined based on the pairwise interaction functions of expression (1) and are presented in the form of a product or minimum or maximum value of the step or Gaussian function for the elements of the point pattern. The benefit of using spatio−temporal models is the ability to reproduce the process of the spread of the disease in dynamics, which allows for a more detailed analysis of its stages.

### 2.3. Geyer Saturation Process

The results of benchmarking the approximation of a spatial random process model describing the spread of tree disease are shown in Table 1. All of these models are realized by the "spatstat" package of the R software environment [30]. Generalization of the Geyer saturation point process model for the case of several interaction distances seems to produce the best result. It can also be described as a saturated equivalent of the process of pairwise interaction with a piecewise constant potential of a pair of points [30], which is listed second in the table based on the value of the Akaike index (2). Significantly, the value of the Akaike index for the inhomogeneous Poisson process model was the largest among the studied processes of tree damage.

**Table 1.** Akaike index values for models of pairwise interaction processes.

| Pairwise Interaction Process | AIC Index |
| --- | :---: |
| Saturated Pairwise Interaction | 800 |
| Piecewise Constant Pairwise Interaction | 1291 |
| Penttinen Interaction | 1544 |
| Baddeley–Geyer | 1648 |
| Geyer | 1649 |
| Multitype Strauss | 1870 |
| Hierarchical Strauss | 1874 |
| Strauss/Hard Core | 1875 |
| Fiksel | 1877 |
| Strauss | 1940 |
| Diggle–Gratton | 1941 |
| Diggle–Gates–Stibbard | 1942 |
| Ord's Interaction | 1954 |
| Area Interaction | 1976 |
| Soft Core | 2217 |
| Lennard–Jones | 2247 |
| Inhomogeneous Poisson | 2297 |

Geyer's saturation processes belong to the class of Gibbs random point process models and are a development of Strauss processes. Due to their structure, they can model a wide range of accidental point configurations from random, when the saturation threshold is equal to zero, to regular ($\gamma < 1$) and cluster ($\gamma > 1$). Thanks to this, they are widely used as models of processes that describe various physical instances such as forest fires, seismic activity and the growth of plantations in the forest.

In the Geyer saturation point process model presented in [30], the function's parameters include the interaction radius (r) and the saturation threshold (s). The interaction radius was determined as the minimum distance to the nearest neighbour among all elements in the point configuration obtained from remote sensing data processing. This choice, known as the measure of spatial relationships [41], was also made based on both the method of spreading the disease through insects and the formation of a point configuration using the centres of image fragments. The saturation threshold value was set at two to maintain the model's ability to form cluster configurations while accounting for the point configuration's formation.

Besides the Akaike index calculation, another way to verify the selected model is to use generalized statistics, such as the K−function, or its derivative, the L−function [30]. However, these are defined only for stationary or at least correlation−stationary models. Residual generalized functions avoid this limitation [32]. The residual K−function for the fitted model is a modified K−function, the mean value of which is zero for the accurate model. It is defined as the difference between the non−parametric estimate of the function K and its model compensator [30].

## 3. Results

The investigated point patterns were obtained based on images of the Shatsk National Nature Park (SNNP), Ukraine, retrieved during the year 2020 from the Landsat8 satellite in the visual spectral range. Images obtained between June and September were selected for the research as those that fully depict the vegetation of the investigated region. The surface of the SNNP is flatland, a plain with a slight slope to the north and absolute heights in the range of 160–180 m above sea level. That allows us not to take into account the factor of the relief height when choosing the model's parameters.

Landsat 8 images contain data from 9 spectral bands at a resolution of 30 m (with the exception of band 8). As shown in [42], bands B2–B5, which contain data on visible (blue, green, red) and near−infrared radiation (0.45–0.88 μm), are important for the study of trees that have been damaged by bark beetles. The Red–Green index (RGI) = B4/B3 and the normalized difference vegetation index (NDVI) = (B5 − B4)/(B5 + B4) are calculated from these bands. These indices allow the differentiation of image elements corresponding to affected trees [42,43] and have been used to create a training set of image patches for two classes—those that contain affected trees and those that contain only healthy trees.

The accuracy of image fragment classification, which depends on the correctness of the training set definition based on the RGI and NDVI, is one of the limitations of the proposed approach. Using unmanned aerial vehicles to collect imagery may seem like an easier way to go when there is no access to high−resolution satellite data.

A classification method based on a convolutional network is applied to a set of patches of the input image [44]. A set of image patch centres of a particular class is considered as a random point configuration while the class labels are used as marks for every point. A marked point pattern is regarded as a union of several sub−point patterns each containing points of the same qualitative marks. It shows the location of affected and healthy trees. Healthy trees are marked with a triangle while infected trees with a circle. Let us consider the original point pattern (Figure 1a) and an example of point pattern simulation using the pairwise interaction process with a piecewise constant potential of a pair of points (Figure 1b) and compare their generalized characteristics.

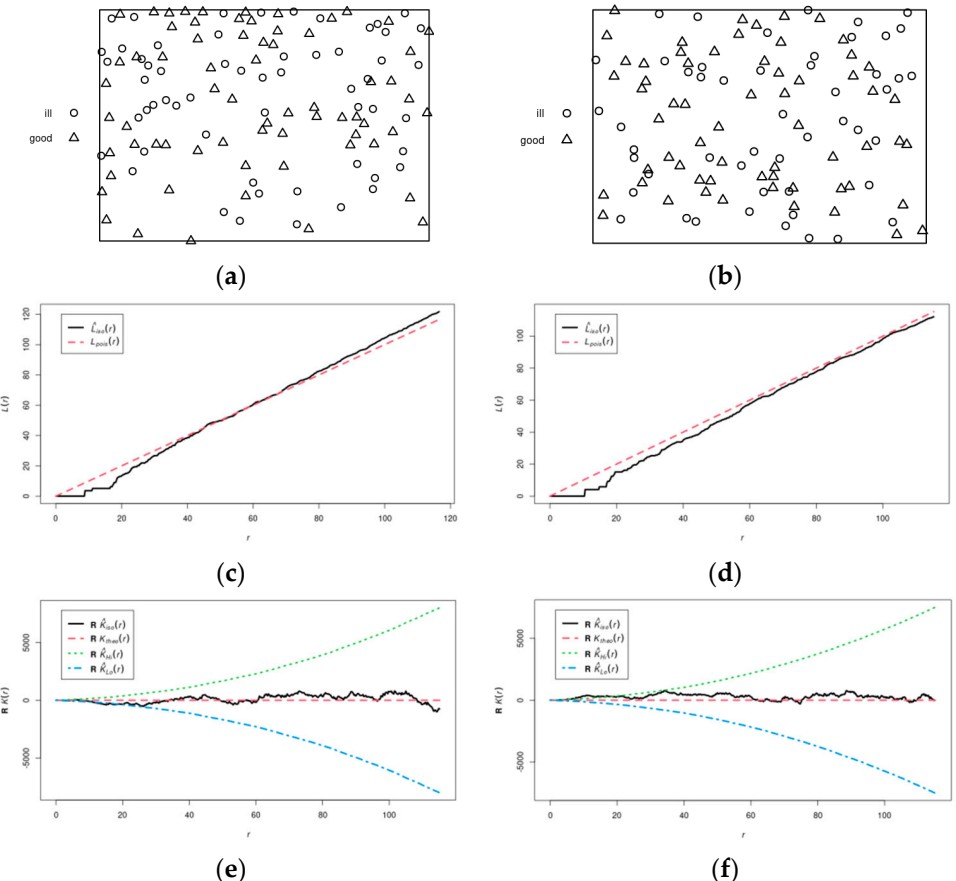

**Figure 1.** Simulation and verification of the contamination process based on point pair interaction. (**a**) Real point pattern; (**b**) simulated point pattern for the generalized model of the Geyer saturation point process in the case of several interaction distances. (**c**) The L−functions of the original process; (**d**) L−functions of the process of pairwise interaction with a piecewise−constant pair potential. (**e**) The residual K−functions for the process of pairwise interaction with a piecewise−constant pair potential; (**f**) the residual K−functions for the generalized model of the Geyer saturation point process in the case of several interaction distances.

Figure 1 also shows L−functions of both the original processes (Figure 1c) and generated ones (Figure 1d), based on the model of pairwise interaction with a piecewise constant potential of the pair. Their comparison suggests a good correspondence of the model data with the real data. Observing the value of the L−function at small distances (less than 20 m) reveals the regularity of the location of the pattern points for this range in both graphs. Further, as interpoint distance increases, a slight trend towards cluster formation of points is observed in both cases.

Let us also consider the values of the residual K−functions for the process of pairwise interaction with a piecewise−constant pair potential (Figure 1e) and the generalized model of the Geyer saturation point process in the case of several interaction distances (Figure 1f). In both cases, for distances up to 20 m, the average value of the function is close to zero, which indicates a good fit of the model to actual data. Furthermore, the mean value of the residual K−function stays close to zero for the entire range of distances between objects. Therefore, the chosen model of the pair interaction process is a good approximation of the real process.

A modified contagious process generation algorithm, as described above, was used to simulate the spatio−temporal spread of tree disease. The research used the same point pattern as for the exclusively spatial model. In the above algorithm, as opposed to the one described in the "stpp" package, the first element of the process was chosen randomly

among the points corresponding to the contaminated trees. At every iteration of the algorithm, the nearest element in the original point pattern was selected as the next infected tree. As a result, infection process was generated as shown in Figure 2a.

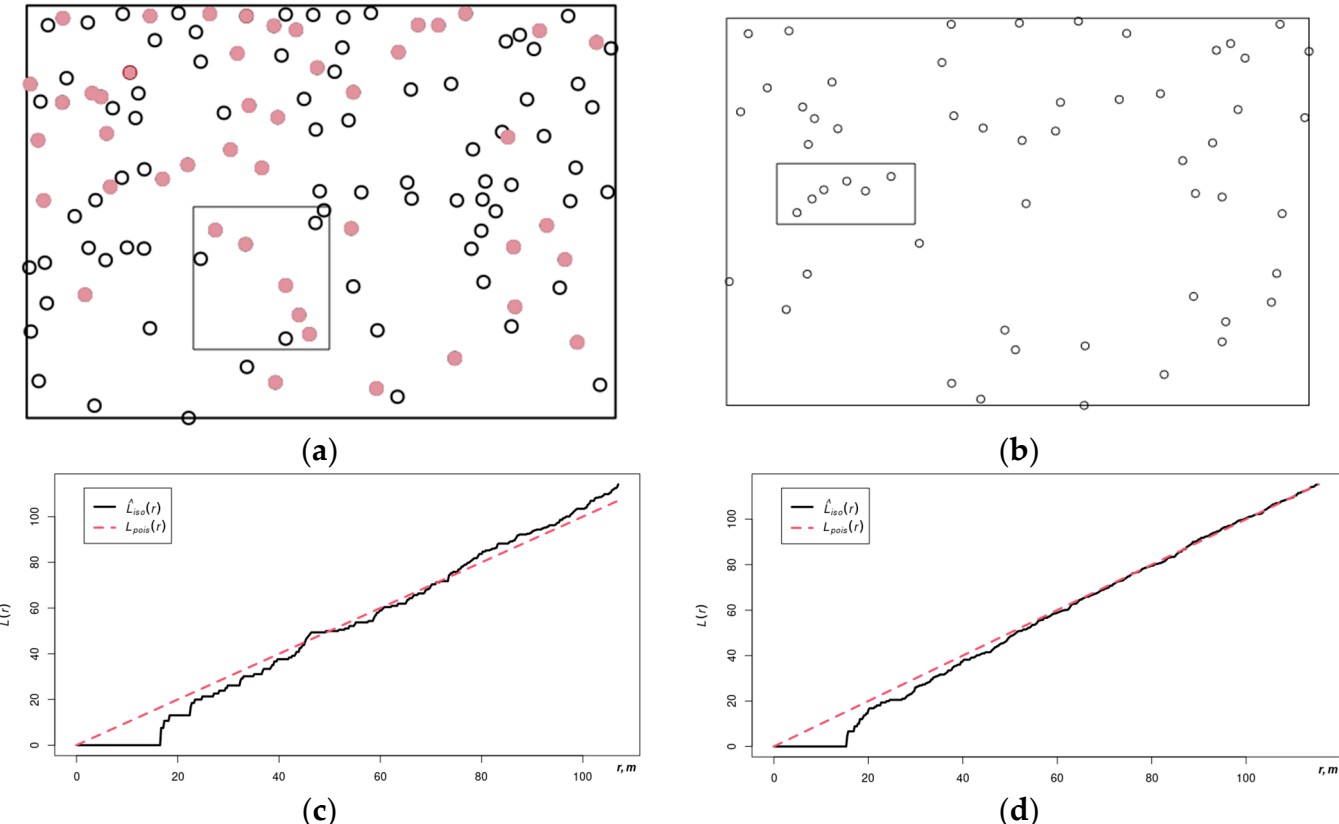

**Figure 2.** Actual and generated point image and their characteristics. (**a**) A point pattern of the actual position of all trees with simulated marks of contaminated trees (red points); (**b**) A point pattern of the actual contaminated trees; (**c**) The L−functions of the simulated positions of contaminated trees; (**d**) L−functions of the point pattern of actual contaminated trees.

Because of the random nature of the model's point pattern generation, there will not be a high degree of visual similarity. One can determine how well the model matches the real data by comparing the features of the pattern generated by the model with the corresponding features of the real data. Analysis of the model−generated (Figure 2a) and original point images of contaminated trees (Figure 2b) shows some similarities in the formation of pictures of infected trees. In particular, point clusters of a similar nature may be found in both point images; the above model creates groups of infected trees known as "curtain groups", as in real conditions [10], which are highlighted by rectangles. For their refined comparison, we calculated the value of the L−function for each of the patterns in the available range of distances. The results are presented in Figure 2c,d. The similarity of the values of the L−function for both point patterns is observed in almost the entire range of distances and suggests good correspondence of the contagious process described in [45] to the real one.

Note that the contagious process has been applied rather than the infection process in modelling the spread of tree damage. The reason for this is a characteristic phenomenon in trees, namely immunity to infection, in this case to a colony of bark beetles. A healthy tree, in the presence of favourable factors, such as sufficient soil moisture and nesting of insectivorous birds and ants, usually copes well with the infection. Moreover, infected but not contaminated trees are almost impossible to distinguish using remote sensing.

Just like in the case discussed earlier, comparing the L−functions of original and contagious processes, generated based on the model of pairwise interaction with piecewise−constant potential of a pair of points, shows good correspondence of model data to the real data. Analysis of the L−function values at small distances (less than 40 m) indicates the regularity of the location of the image points in both graphs. As inter−point distance is increased, a slight trend towards cluster formation of points is observed. Thus, pairwise interaction with piecewise constant pair potential can be used as an adequate model of the pine disease spread by bark beetles.

## 4. Discussion

Among many factors affecting the resistance of trees to diseases are increasing temperature and decreasing groundwater level, caused by climate change [13,20]. Conditions are being created that are favourable for the development of pests but harmful for trees with a simple root system. Thus, the process of mass death of pine forests due to the action of bark–tracheomycosis complexes is rapidly developing [39].

The drying of trees of one or more species, especially at different stages of ontogenesis, is a process lasting in time and space, and depends on many factors. Since forest ecosystems are the most complex biological complexes in the organic world, their pathological processes constantly interact with many organisms of different taxonomic groups [1,2]. It is important to understand the scale and consequences of these processes to be able to preserve the plantation population. In the case of an individual tree, the causes of the disease are relatively easy to explain, and the patterns that describe its course are known [12]. The spread of the disease and its speed depend on numerous environmental factors, and are often unknown. Hence, modelling the decease progression is vital for their assessment. There are known models that provide a quantitative indicator of the loss of trees in terms of mass or volume [2,3,6]. The spatial distribution of the disease, however, is not being considered. Omitting the spatial factor, in our opinion, is a significant shortcoming, as it disregards or simplifies additional aspects of the mutual influence between the disease and the environment.

Recently, a number of studies have been proposed [17–19,23,42,43] for the investigation of pine tree diseases on the basis of the analysis of remote sensing images. In the vast majority of cases, this analysis is limited to the application or improvement of image classification technology with the help of deep learning networks. This makes it possible to improve the accuracy of the identification of infected trees, but it does little to explain the evolution of the disease. Here, we propose to complement this analysis with models of disease evolution.

Spatial and spatio−temporal models of the tree disease spread are considered in the paper, based on random point processes. It is shown that the spreading process of pine drying due to damage caused by bark beetles can be modelled using random point processes of pair interactions with piecewise constant pair potential. Spatial random point processes are convenient for modelling the phenomena that resemble spontaneous events occurring at specific locations [31,32,34,35,37]. They allow for both evaluating the characteristics of the physical processes taking place and establishing the possible connection between different processes.

Spatial point processes are seldom used for modelling the spread of tree disease. The reason for this, in our opinion, is that the locations of trees in some areas are fixed, and so, as mentioned in [29], the spread of the disease cannot be considered a random point process. We have investigated the possibility of using random point processes to model tree diseases by generalizing the formation of point patterns [44]. A point image element is the centre of a fragment of a remote−sensing image that corresponds to a given criterion. Such a fragment contains a random number of infected trees. Also, matching the criterion of the spectral characteristic by the fragment is of a random nature. This way, the problem of the fixed trees has been (to a large degree) resolved.

Spatial random processes are also widely used to describe contagion and infection processes in humans or animals [21,45]. We suggest that it is possible to describe the process of tree disease based on models of the contagion process. Proposed models have been successfully verified, thus confirming the validity of our assumption. This allows us to use already (well) established apparatus to describe the process of tree contamination. Due to similarities in their structure, the spatial process with pairwise interaction and spatio−temporal contagious process produce similar verification results. The spatio−temporal contagious process is a spatial process with interaction that takes into account the time component, allowing to identify the moment of occurrence of each individual event [40].

It is not realistic to account for the large number of various factors when building a model, as this will result in an overly complicated model. Using models based on random point processes and their developed apparatus allows for a description of the natural processes. The advantage of spatial modelling is that it allows us to reconstruct a realistic picture of the spread of the disease despite accounting for generally small fraction of factors that influence the course of the disease. For instance, paper [11] examines the progression of drying of pine trees due to damage caused by bark beetles and analyses the impact of trees brought down by a storm. Even though both sets of trees are considered point patterns, the authors do not examine the mutual influence between these images using, for example, the marker correlation function or the marker connection function [30].

Validation of the model is a crucial stage of checking the correctness of the hypothesis embedded in it to describe the real world. The simulation model results in comparison with the results obtained in authentic conditions allow us to confirm the correctness of the model [36]. The use of models is justified in many cases, for example, when conducting a physical experiment is too expensive or dangerous. Another instance is when the model is designed to predict true events occurring over a long period. This enables to save time and prevent losses if they are foreseen. Nevertheless, the complete validation of the model in such cases is complicated, because it means assuming the occurrence of such losses. Therefore, the development of this kind of model can be used primarily to assess the evolution of the studied natural phenomenon. For example, models for forest fire expansion can be laborious to validate on actual data, because in most cases, actions are taken to prevent the fire from spreading [31,32]. Of course, some phenomena are difficult to avoid; typically, these are spontaneous natural phenomena such as earthquakes, floods and storms, for which it is realistic to check the simulation results against real data. However, the problem considered herein of pine trees drying cannot be attributed entirely to their numbers because, as in the case of fires, precautionary measures are applied to reduce disease impact. Given that random point processes generate random configurations of points, the results obtained by applying the models considered are of limited use. Although it is possible to estimate the area where the disease is spreading, e.g., as the area of the envelope of a polygon, the quantitative loss of wood can be estimated very approximately.

## 5. Conclusions

Much effort is being made to explain, predict and eliminate contagion in pine plantations. Accurate modelling of disease development makes it possible to take preventive measures over time, thereby protecting trees from destruction. Spatial models that depict the spread of forest infections have been a recent focus, in addition to modelling individual tree diseases. Such models make it possible to use data from field research as well as remote sensing data of the forest surface affected by the disease. Using remotely sensed data also makes it possible to automate and greatly simplify the process of surveying large areas of forest plantations.

The use of spatial point processes has been widely used to study forest stands. Its main application is in the description of the development of plantations, taking into account the mutual relations between them. There are a number of models of random point processes that have been proposed to describe the development of trees in a forest stand. On the basis of the above arguments, we investigated the application of random point process models

to describe the spread of tree diseases, in particular in pine plantations damaged by the apex bark beetle. It was found that models based on saturation processes are best suited to the problem under study, according to the results of the research conducted.

The use of spatio−temporal models to describe the contagious processes in forest plantations allows us to estimate the rate at which the disease will spread and to predict the damage it will cause in real time.

We have shown that the process of analysing the condition of forest stands can be significantly accelerated by using remote sensing data. In particular, by generating random point configurations based on remote sensing data and testing them against certain random process models, it is possible, for example, to distinguish between tree damage caused by storms and damage caused by disease.

**Author Contributions:** Conceptualization, R.K.; Methodology, I.J.-K.; Software, O.L.; Validation, O.L.; Formal analysis, I.J.-K.; Investigation, R.K. and O.L.; Writing—original draft, R.K.; Writing—review & editing, A.S.; Supervision, B.R.; Project administration, B.R.; Funding acquisition, A.S. All authors have read and agreed to the published version of the manuscript.

**Funding:** This research was funded by the Silesian University of Technology, statutory research no. BK-274/ROZ1/2023 (13/010/BK_23/0072).

**Data Availability Statement:** The data presented in this study are available on request from the corresponding author.

**Conflicts of Interest:** The authors declare no conflict of interest.

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
