# Peer review of "Analysing Pine Disease Spread Using Random Point Process by Remote Sensing of a Forest Stand"

_remotesensing, doi:10.3390/rs15163941_

Round 1

Reviewer 1 Report

This article addresses an interesting approximation to model the process of trees infestation, considering spatial and spatio-temporal models and diverse kinds of models useful in other issues such as forest fire.

The introduction is correct, and the materials and methods are clear enough, even though the authors facilitate information about methods in the results section (lines 242 -274 and 297–303), increasing it with information which should be described in methods.

Results should be improved, in particular the Figure 2 that supports the explanation and results. The similarity between actual and simulated point models is unclear, and the quality of the legend does not permit to distinguish between different information.

The discussion needs to be supported by more works, e.g., the text in lines 343-348 or 368-374 and may be focused on the implications to use this method in a more general situation.

The authors could revise the conclusions because some of them are not exactly conclusions, but summaries of the method.

Hopefully, my comments will help the authors.

Author Response

Comments from Reviewer 1

Comment 1: Results should be improved, in particular the Figure 2 that supports the explanation and results. The similarity between actual and simulated point models is unclear, and the quality of the legend does not permit to distinguish between different information.

Response: We agree with this comment. Therefore, we have made changes to Figure 2, as well as slightly modified the text of the article related to the description of this figure (lines 336-346).We also made changes to the legends in Figure 1.

Comment 2: The discussion needs to be supported by more works, e.g., the text in lines 343-348 or 368-374 and may be focused on the implications to use this method in a more general situation.

Response: Agree. We have amended the Discussion section to include a reference to the related work as follows.

Comment 3: The authors could revise the conclusions because some of them are not exactly conclusions, but summaries of the method.

Response: We agree with this and have completely revised the Conclusions so that they do not include a summary of the method. (lines 449-473).

Reviewer 2 Report

ANALYZING OF PINE DISEASE SPREAD USING RANDOM 2 POINT PROCESS BY REMOTE SENSING OF FOREST STAND.

1.       Authors need to write research questions and hypothesis for this research

2.       Does the author use remote sensing indexes to help diseased pines, given that this study employs a remote sensing method? Explain the remote sensing index used, as well as its calculation and the state of the art associated with the selected remote sensing index.

3.       How to determine indices value based on the remote sensing index used to evaluate healthy pine and diseased pine?

4.       Since this research using data from Landsat 8, authors need to write band sensor for Landsat 8 used

5.       Several similar studies have already been conducted; writers must describe the contribution and strength of this study in comparison to earlier studies.

6.       Authors need to write limitation of this research

Author Response

Comments from Reviewer 2

Comment 1: Authors need to write research questions and hypothesis for this research.

Response: Thank you for pointing this out.We hope that we've sufficiently stated the aims of our paper in the Introduction, where we set out the direction in which we wanted to go, based on what was known in the field. However, for the sake of clarity, we have added a statement that forms the basis of the research we have undertaken. (lines 91-93)

Comment 2: Does the author use remote sensing indexes to help diseased pines, given that this study employs a remote sensing method? Explain the remote sensing index used, as well as its calculation and the state of the art associated with the selected remote sensing index.

Response: You are absolutely right to draw our attention to these issues with remote sensing indices, which we had overlooked. In order to classify the input image patches and form point patterns, we used a common RGI and NDVI indices to form the train sequences of image patches.

Comment 3: How to determine indices value based on the remote sensing index used to evaluate healthy pine and diseased pine? 

Response: From the literature survey we found that the aforementioned RGI and NDVI indices are the most informative and practically applicable for the identification of image regions that correspond to dead trees.

Comment  4. Since this research using data from Landsat 8, authors need to write band sensor for Landsat 8 used.

Response: Agree. We have added a description of the band ranges we use in our research. (lines 263-271)

Comment 5.  Several similar studies have already been conducted; writers must describe the contribution and strength of this study in comparison to earlier studies.

Response: Agree. In the Discussion section, we have described some of the limitations that we believe are inherent in the use of remote sensing for forest research. (lines 381-387)

Comment 6   Authors need to write limitation of this research.

Response: Agree. We have added a number of arguments describing the limitations of our approach. (lines 272-275; 442-446)

Reviewer 3 Report

In the article "Analyzing pine disease spread using random point process by remote sensing of forest stand," the authors employ an approach to generate random point configurations in order to describe the occurrence of plant diseases. Unfortunately, the article needs significant revision before being published in Remote Sensing as it lacks substantial relevance to the journal's topic. Additionally, the article requires restructuring as it primarily focuses on theoretical aspects without demonstrating their practical applicability. There is a lack of discussion on remote sensing-related themes in the article.

Author Response

Comments from Reviewer 3

Comment 1:  Unfortunately, the article needs significant revision before being published in Remote Sensing as it lacks substantial relevance to the journal's topic.

Response: Thank you for this suggestion. It was our understanding that the title of the special issue “Remote Sensing Applications for Forest Ecosystem Monitoring and Spatial Modeling” was intended to cover research that is developing or refining methods for the analysis of remote sensing data describing the state of forest ecosystems. Our research relies heavily on remote sensing data and provides a tool for ecosystem monitoring based on spatial modelling. That is why we have submitted the article to this special issue.

Comment 2: Additionally, the article requires restructuring as it primarily focuses on theoretical aspects without demonstrating their practical applicability. There is a lack of discussion on remote sensing-related themes in the article.

Response: You have raised an important point here. We are fully in agreement that the practical application of remote sensing methods is an important issue. It should not be limited to the development or improvement of data processing methods, such as the development of neural network architectures for the classification of remote sensing images. Practical applications should be more complex, such as the explanation of the real situation observed in the area under study. Taking into account other comments, we have completed the article with our vision of the practical application of the proposed approach to the problem of forest monitoring.

Round 2

Reviewer 3 Report

The authors made changes in the article and I no longer have any objections.